# Holoprosencephaly with a Special Form of Anophthalmia Result from Experimental Induction of *bmp4*, Oversaturating BMP Antagonists in Zebrafish

**DOI:** 10.3390/ijms24098052

**Published:** 2023-04-29

**Authors:** Johannes Bulk, Valentyn Kyrychenko, Philipp M. Rensinghoff, Zahra Ghaderi Ardekani, Stephan Heermann

**Affiliations:** Department of Molecular Embryology, Institute of Anatomy and Cell Biology, Faculty of Medicine, University Freiburg, 79104 Freiburg, Germany

**Keywords:** Bone Morphogenetic Protein (BMP), antagonists to Bone Morphogenetic Protein (BMP antagonists), eye field, forebrain, holoprosencephaly, *rx2*, *rx3*, crypt-oculoid

## Abstract

Vision is likely our most prominent sense and a correct development of the eye is at its basis. Early eye development is tightly connected to the development of the forebrain. A single eye field and the prospective telencephalon are situated within the anterior neural plate (ANP). During normal development, both domains are split and consecutively, two optic vesicles and two telencephalic lobes emerge. If this process is hampered, the domains remain condensed at the midline. The resulting developmental disorder is termed holoprosencephaly (HPE). The typical ocular finding associated with intense forms of HPE is cyclopia. However, also anophthalmia and coloboma can be associated with HPE. Here, we report that a correct balance of Bone morphogenetic proteins (BMPs) and their antagonists are important for forebrain and eye field cleavage. Experimental induction of a BMP ligand results in a severe form of HPE showing anophthalmia. We identified a dysmorphic forebrain containing retinal progenitors, which we termed crypt-oculoid. Optic vesicle evagination is impaired due to a loss of *rx3* and, consecutively, of *cxcr4a*. Our data further suggest that the subduction of prospective hypothalamic cells during neurulation and neural keel formation is affected by the induction of a BMP ligand.

## 1. Introduction

Vision is likely our most prominent sense. A correct development of the eye is at its basis. Overall, eye development is an intricate yet fascinating process. During gastrulation, a single eye field within the anterior neural plate (ANP) is divided and, subsequently, two optic vesicles emerge at the lateral surface of the anterior neuroectoderm. The progenitors within the eye field behave like single cells [1] and migrate differently compared to the telencephalic progenitors [2]. During neurulation the telencephalic progenitors converge towards the midline while the eye field progenitors are aligned more laterally and move even further away from the midline [1]. This out-pocketing of optic vesicles is dependent on *rx3* and *rx3*-dependent expression of *cxcr4* [3]. Importantly, the division of the ANP is also supported by a subduction movement of hypothalamic progenitors occurring during neurulation and neural keel formation. These progenitors are located posterior to the eye field originally but then migrate ventrally and rostrally inducing neural keel formation. This movement indirectly supports the division of the ANP [4]. However, not only must the eye field be divided, but the prospective telencephalic domain within the ANP must also split, resulting in two telencephalic lobes. Failures during these processes cause holoprosencephaly (HPE). The spectrum of HPE phenotypes varies between mild forms and intense forms including the classical ocular phenotype cyclopia, but in other cases also anophthalmia or coloboma [5]. HPE is mostly genetically linked. The most prominent factor affecting ANP division is the Sonic hedgehog protein (Shh), derived from the prechordal plate [6,7]. BMP signaling must be activated at a specific level during the initiation of the ANP to facilitate the expression of Wnt antagonists in the anterior neural border [8] which in combination with Wnt ligand expression in posterior domains ensure a functional Wnt gradient over the ANP [9]. This is important for ANP domain specification. BMP signaling was found to be favoring the telencephalic progenitor fate over the fate of eye field progenitors [3]. BMP signaling can be affected by different means, e.g., by regulation of ligand or receptor expression but also by induction of BMP antagonists. The latter are frequently found expressed redundantly, underlining their pivotal role during embryogenesis [10,11]. BMP antagonists, derived from the mouse node and axial mesendoderm, are essential for head development. A combined loss of chordin and noggin affects the formation of the prosencephalon and can also result in cyclopia [10], thus providing embryos with variable intense phenotypes in the HPE spectrum. A loss of BMP antagonists will result in an overactivation of BMP signaling. Nevertheless, an enlargement of the telencephalon at the expense of the eye field as seen after BMP signaling activation in zebrafish [3] was not described resulting from the loss of BMP antagonists in mice. This may be due to species differences [10].

As mentioned above, HPE can also be associated with a coloboma. We previously described a coloboma resulting from hampered morphogenetic movements during optic cup formation rather than from a fusion defect [12,13]. We found a BMP antagonist expressed at sites of high cellular mobility and showed that an induction of a BMP ligand (*bmp4*) is capable of halting optic cup morphogenesis [12,13,14]. We reasoned that such a “morphogenetic coloboma” is more likely linked to HPE than a coloboma resulting from a fusion defect of the optic fissure margins, happening significantly later during development, and asked if BMP antagonism is also important for proper forebrain cleavage and if an induction of a BMP ligand is potentially hampering early forebrain development.

In this study we thus addressed the role of BMP antagonism during ANP and eye field development in zebrafish (*Danio rerio*). We found BMP antagonists, *fsta*, *chrd*, *nog2* and *grem2b*, expressed in the ANP at 11 hpf (hours post fertilization) and 12 hpf. To oversaturate the BMP antagonists in this domain, we experimentally induced *bmp4* expression at 8.5 hpf. This induction resulted in anophthalmia. Notably, however, we found *rx2* and *pax6* positive retinal progenitor cells being stuck in the forebrain. Expression analysis of markers within the ANP at 11 hpf showed that the division of the telencephalic field and the division of the eye field were hampered. We also found a loss of *rx3* and *cxcr4* expression, a reduced expression of *shha/b* in the prechordal plate and a ceased *zic2* expression in the ANP. Further analyses suggest a failure of neural keel formation including an impaired hypothalamic subduction movement during neurulation.

## 2. Results

### 2.1. Bmp4 Induction at 8.5 hpf Results in HPE and a “Crypt-Oculoid”

We investigated whether BMP antagonists are important for the division of the ANP including the eye field. To this end, we first addressed the expression of BMP antagonists in the ANP. We found BMP antagonists *fsta*, *chrd*, *nog2* and *grem2b* expressed in the region of the ANP at 11 hpf and 12 hpf (Figure 1A–Q, dotted lines and arrows). The expression of *nog2* was broader at 11 hpf and condensed to the midline at 12 hpf. Subsequently, it did no longer reached the most anterior domain of the embryo (Figure 1A–D, dotted lines and arrows). The expression of *chrd* was weak and found along the midline at 11 hpf and was almost absent at 12 hpf (Figure 1E–H, dotted lines and arrows). At 11 hpf the expression of *fsta* was found weak along the midline of the body axis while the expression at 12 hpf was found bulging in the anterior domain reminiscent of the out-pocketing optic vesicles (Figure 1I–M, dotted lines and arrows). The expression of *grem2b* was also found along the midline. However, an onset of bulging in the area of the future optic vesicles was detectable already at 11 hpf and was increased at 12 hpf (Figure 1N–Q, dotted lines and arrows).

We next aimed to oversaturate these expression domains in the ANP region using a transgenic line allowing a heat-shock-induced expression of *bmp4* (*tg(hsp70l:bmp4, cmlc2:GFP)*) mated with wildtype zebrafish. We were aware that this approach would not allow a precise spatial induction, yet the timing of induction is well controllable. We have shown previously that an induction of *bmp4* is inducing BMP signaling [12,14,15]. Here, we performed *bmp4* induction at 8.5 hpf (Figure 1R). First, we showed that embryos of the *tg*(*hsp70l:bmp4, cmlc2:GFP)*) are responsive to an induction at 8.5 hpf by using a BMP reporter line (Appendix A). At this developmental stage the fluorescent reporter (*cmlc2:GFP*) indicating the inducible *bmp4* was not yet visible. Therefore, many embryos of several clutches were subjected to a heat shock. We next performed a gross morphological analysis at 48 hpf using a stereomicroscope. At this age, the embryos containing the transgene could be identified by the transgenesis marker (*cmlc2:GFP*). Heat shocked embryos without the transgene served as controls (36 embryos) (Figure 1S,T). In *bmp4*-induced embryos, we found a dysmorphic forebrain and could not detect any eyes in 97.5% of the analyzed embryos (39 out of 40). In addition, we observed a pericardial edema and a curved tail (Figure 1U,V).

We next asked whether residual “retinal tissue” or retinal precursors could exist inside the dysmorphic forebrain. To test this, we used another transgenic zebrafish line *tg(rx2:GFPcaax)* [13], in which retinal progenitors are expressing GFP, localized to membranes. We crossed fish from *tg(rx2:GFPcaax)* and fish from *tg(hsp70l:bmp4, cmlc2:GFP)*, injected RNA coding for LY-tdtomato and subjected the offspring to a heat shock at 8.5 hpf. We used confocal microscopy for analysis of the embryos at 24 hpf. At this age the transgenesis marker of *tg(hsp70l:bmp4, cmlc2:GFP)* was visible, enabling separation of *bmp4*-induced from un-induced embryos. The latter served as controls. Embryos showing GFP expression in the optic cups or inside the forebrain were analyzed with a confocal microscope. In controls (eight embryos), GFP expression was found regularly in retinal progenitors inside the optic cups (Figure 1W). In *bmp4*-induced embryos (20 embryos), we found many GFP expressing cells inside the forebrain. These cells were found close to the midline and no clear separation into left and right domains could be distinguished. Furthermore, no optic cups could be found (Figure 1X).

Besides the GFP expression driven by the *rx2* cis regulatory element, we also addressed the endogenous *rx2* transcript in controls (five embryos) and *bmp4*-induced embryos (five) using whole mount in situ hybridization (WMISH). The WMISH showed *rx2* transcript expression corresponding nicely to the GFP expression driven by the *rx2* cis regulatory element (Figure 1Y–Bb). Together this indicates that induction of *bmp4* at 8.5 hpf is hampering eye field splitting and optic vesicle out-pocketing, since no eye was formed, yet retinal precursors were identified inside the forebrain. We termed this phenotype “crypt-oculoid” (Figure 1X,Aa,Bb) as a specific form of anophthalmia.

### 2.2. ANP Development Is Severely Hampered after bmp4 Induction at 8.5 hpf

Having observed that *bmp4* induction hampers eye field splitting, we next addressed the effect of *bmp4* induction on the development of the ANP. To this end, we induced *bmp4* by heat shock at 8.5 hpf using *tg(hsp70l:bmp4, cmlc2:GFP)* fish mated with wildtype zebrafish and fixed the embryos at 11 hpf. Consecutively, we processed the embryos for WMISH (Figure 2A). We used markers for the prospective forebrain (*emx3* and *foxg1*) domain and the prospective eye field (*six3b* and *rx3*). The transgenesis marker (*cmlc2:GFP*) was not yet active at 11 hpf. Thus, the WMISH was performed blinded and the genotype (wt vs. *hsp70l:bmp4, cmlc2:GFP*) was determined after the acquisition of the expression patterns. After genotyping the recorded expression patterns were grouped in “control” and “*bmp4*-induced” samples (Figure 2, wt vs. *hs:bmp4*). Based on the literature it was expected that an excess of a BMP ligand is potentially resulting in an expansion of the prospective telencephalic domain at the expense of the eye field [3]. Nevertheless, in our paradigm we could not detect a shift in fate between these two domains. We rather found that, resulting from the induction of *bmp4*, the forebrain domain (*emx3*) and the eye field (*six3b*) were both condensed at the midline (Figure 2B–E arrows, Figure 2K–N/4; *bmp4*-induced embryos and four control embryos were analyzed, respectively). This indicates that the induction of *bmp4* was hampering the division of both the eye field and the prospective forebrain domain, being the hallmark of HPE. Interestingly, the expression of *rx3* and *foxg1a* were ceased/lost after *bmp4* induction (Figure 2F–I,O–R arrows/4 *bmp4*-induced embryos and four control embryos were analyzed, respectively). This suggests that either the induction or the maintenance of the expression of these two transcription factors was sensitive to elevated levels of *bmp4*. *Foxg1* is an essential transcription factor involved in many aspects of telencephalic development including also the growth of this brain region [16]. *Foxg1* was also shown to be essential for the development of the olfactory system including the olfactory epithelium and also the olfactory bulb, a telencephalic region [17]. A loss of *foxg1* expression could thus likely be resulting in reduced size of the telencephalon. A loss of *rx3* could be a plausible reason for the anophthalmia phenotype [1,2,3,18].

### 2.3. Bmp4 Induction Results in Loss of rx3 Expression but Sustained Expression of rx2

We found it surprising, however, that *rx3* expression was lost after *bmp4* induction while *rx2* positive progenitors could still be detected at a later stage of development (Figure 1Y–Bb and Figure 2O–R). In the zebrafish *rx3* mutant *chokh*, it was shown that *rx2* expression depends on *rx3* expression [18]. In medaka, however, the expression of *rx2* is sustained when *rx3* is lost [19]. We next targeted the *rx3* locus of the zebrafish genome with CRISPR/Cas9 and analyzed F0 Crispants (Figure 3A), using the protocol of Wu and colleagues [20]. The specific sequences chosen for sgRNA preparation are given in the materials and methods section. 118 zygotes were injected. Of these, 59% (70) developed an anophthalmic phenotype. Twenty embryos (17%) showed normal eyes, while the remaining embryos showed differential ocular phenotypes such as bilateral microphthalmia (16), a single lateral eye (six), dysmorphic eyes (three) and unilateral microphthalmia (three). Thus, we state that we reproducibly observed anophthalmia and severe microphthalmia in our F0 Crispants (Figure 3B–E). We next analyzed the *rx2* expression in F0 *rx3* Crispants by WMISH (Figure 3F–I). Notably, we found severely reduced to absent expression levels of *rx2* in our F0 *rx3* Crispants (strong phenotype), correlating nicely with the efficacy of the CRISPR-induced anophthalmia phenotype (nine embryos). This supports the finding that *rx2* is dependent on *rx3* expression in zebrafish. However, this finding does not explain why *rx3* is absent after *bmp4* induction, while *rx2* expression is present afterwards (Figure 1Aa,Bb and Figure 2Q,R). Overall, our findings show two different forms of anophthalmia. We will discuss this alleged contradiction between these below.

### 2.4. Bmp4 Induction Alters Expression of zic2a, shha/b, alcamb and cxcr4a

Next, we addressed factors influencing forebrain and eye field splitting. We performed this analysis at 11 hpf, subsequent to a heat shock at 8.5 hpf, on embryos derived from mating *tg(hsp70l:bmp4, cmlc2:GFP)* with wildtype zebrafish and the experimental design as aforementioned. The most prominent cause of HPE is the loss of Shh secreted from the prechordal plate. We thus addressed the expression of *shha/b* in controls and *bmp4*-induced embryos (Figure 4A–I). In controls, we found *shha* (four embryos) and *shhb* (three embryos) expressed in the axial mesendoderm (Figure 4B,C,F,G). The expression of *shha/b* was reaching far to the anterior end of the embryo, where the domain was broadened and showing strong expression, (arrows in Figure 4B,C,F,G) corresponding to the prechordal plate. After induction of *bmp4*, we found the expression of both *shha* (11 embryos) and *shhb* (nine embryos with strong repression, two embryos with wildtype-like expression) variably reduced (Figure 4D,E,H,I), yet not absent. Notably, the anterior domain of *shha* was less broad and less intense, suggesting an affected prechordal plate (Figure 4D,E arrows).

The transcription factor *zic2* also is an established HPE-related gene. If mutated it hampers prechordal plate development [21] and thus results in HPE. Later in development, *zic2* was, however, also suggested to act downstream of Shh, limiting the expression of *six3* in the developing forebrain [22]. We thus next addressed *zic2a* expression in controls and after *bmp4* induction (Figure 4K–N). While in controls (four embryos) *zic2a* is expressed at the border of the ANP among other domains, we found *zic2a* almost absent in this region after *bmp4* induction (five embryos) (Figure 4M,N).

Next, we addressed the expression of the chemokine receptor *cxcr4a*, which was shown to be regulated by *rx3* [3] and, in combination with *sdf1b* from the underlying mesendoderm, is supposed to be important for eye field evagination. We found a strongly reduced level of *cxcr4a* expression in the eye field after induction of *bmp4* (four embryos) compared to controls (four embryos) (Figure 4O–R). This is in line with previous findings [3] and supporting the idea that the loss of *rx3* and consecutively the loss of *cxcr4a* is a reason for anophthalmia resulting from induction of *bmp4*.

### 2.5. Bmp4 Induction Hampers Hypothalamic Subduction during Neurulation and Neural Keel Formation

As abovementioned, a proper interaction of the prechordal plate and the ANP via Shh is crucial for normal forebrain cleavage [23,24]. Consecutively, during normal neurulation a part of the future hypothalamic domain is subducting and moving in an anterior direction underneath the rest of the ANP [4]. This is an important aspect of neural keel formation which is also increasing the height along the dorso-ventral axis of the head.

We addressed the arrangement of the forebrain at 24 hpf by expression of marker genes for the telencephalic precursors, retinal precursors and hypothalamic precursors, thereby aiming at understanding the effect of an induction of *bmp4* on the hypothalamic subduction during early forebrain development. To this end, we subjected embryos of *tg(hsp70l:bmp4, cmlc2:GFP)* crossed to wildtype zebrafish to a heat shock at 8.5 hpf. At 24 hpf the embryos were sorted in *cmlc2:GFP*-negative (wildtype/controls) and *cmlc2:GFP*-positive (*bmp4*-induced). Consecutively, the embryos were fixed and processed for WMISH.

First, we addressed the expression of *pax6a* and *pax2a*, two important transcription factors for eye development. *Pax2* is important for proximal fates and optic stalk development and *pax6* is important for optic vesicle/cup development [25] (Figure 5B–I). While in controls (six embryos) we found *pax6a* expressed in the optic cups and diencephalon, we found it expressed in the domain of the crypt-oculoid and a domain posterior to this, likely corresponding to the diencephalic domain, after *bmp4* induction (six embryos) (Figure 5B–E, dotted lines). *Pax2a* expression was detected in the optic stalk and the midbrain-hindbrain boundary in controls (six embryos) (Figure 5F,G, dotted lines). After *bmp4* induction, the expression in the midbrain hindbrain boundary was still visible, while optic stalks could not be identified (five embryos) (dotted line Figure 5H,I). Together, our data indicate that the induction of *bmp4* results in a failure of eye field separation and optic vesicle out-pocketing. It also showed that even though retinal precursors (positive for *rx2* and *pax6a*) were present, no eye was formed.

Next, we addressed expression of *emx3* at 24 hpf. *Emx3* is a marker for the prospective telencephalon at early stages (11 hpf), but is also expressed within the diencephalon/hypothalamus at later embryonic stages [26,27]. In controls (five embryos) *emx3* mainly shows expression in the developing telencephalon and mild expression in the anterior ventral diencephalon (Figure 5K,L dotted line and arrows). After *bmp4* induction (five embryos), *emx3* is broadly expressed caudally to the crypt-oculoid. Compared to the control, only a small *emx3*-positive domain is visible in the residual anterior region, also entangling the crypt-oculoid (Figure 5M,N dotted line and arrows).

The localization of the vast *emx3*-positive domain posterior to the crypt-oculoid suggests a failure of the hypothalamic subduction movement. Moreover, the height of the dorso-ventral axis is reduced after *bmp4* induction resulting in a more linear arrangement of the forebrain domains. The vast *emx3*-expressing domain posterior to the crypt-oculoid likely does not solely correspond to the ventral anterior hypothalamic domain in controls. It cannot be ruled out that telencephalic precursors are also misplaced to this domain (Figure 5N right arrow).

We next addressed the expression of other genes marking in part diencephalic/hypothalamic identity. We addressed the expression of *her13* and *fgf8a* (Figure 5O–V). In controls (four embryos) *her13* is expressed in telencephalic and anterior ventral diencephalic regions (Figure 5O,P). After *bmp4* induction (three embryos), however, *her13* expression ceased, especially in anterior regions (Figure 5Q,R). *Fgf8a* is expressed in the midbrain/hindbrain boundary and in anterior regions, including the ventral anterior diencephalon in controls (five embryos) (Figure 5S,T arrows). After induction of *bmp4* (three embryos) the anterior ventral expression is ceased, while two expression domains posterior to the crypt-oculoid could be detected (Figure 5U,V arrows), indicative for a subduction defect of the hypothalamic precursors.

Taken together, induced expression of *bmp4* resulted in a linear arrangement of forebrain domains and a reduced dorso-ventral height of the forebrain, and hypothalamic precursors which did not subduct to the anterior ventral diencephalic position, both indicative of a hampered neural keel formation.

## 3. Discussion

Holoprosencephaly (HPE) is the most frequent developmental forebrain disorder in humans. The incidence in live births is approximately 1/10000. However, the estimated incidence per conception is much higher, 0.4% determined in abortions [28]. The intensity of the HPE phenotype can be variably pronounced as well as the ocular phenotypes observed concomitantly [29]. Cyclopia, anophthalmia but also coloboma can be found [5,29].

In this study (Figure 6, scheme, summary of major findings) we set out to address the role of BMP antagonism during early forebrain and eye development. We found different BMP antagonists (*grem2b, chrd, nog2* and *fsta*) expressed in the region of the ANP at 11 hpf and 12 hpf. We have previously shown that an induction of *bmp4* can induce BMP signaling in domains of BMP antagonists [12,13] and halt morphogenetic movements during eye development.

Here, we induced *bmp4* at 8.5 hpf and found a severe form of HPE associated with anophthalmia in 97.5% of the cases, indicating a robust experimental paradigm. In mice, a compound knock-out of two BMP antagonists, chordin and noggin, resulted in severe forebrain defects presenting cases of HPE with cyclopia but also more severe cases with aprosencephaly [10]. Even though the phenotypes of the compound mouse mutants and ours, induced by experimental *bmp4* induction, were both showing severe HPE, there were also marked differences. Importantly, we found anophthalmia with crypt-oculoids instead of cyclopia.

Our analyses of the ANP at 11 hpf revealed a splitting defect of the eye field and the prospective telencephalon. We found markers for the future telencephalon (*emx3*) and the eye field (*six3b)* condensed at the midline. The proper interaction of the prechordal plate mesoderm with the developing prosencephalon, the ANP at this stage, is likely the most important step for normal cleavage of the telencephalic field and the eye field [23]. Most HPE-related genes and also non-genetic risk factors can be linked to this step [30]. Shh is the essential factor secreted from the prechordal plate [24] and in turn is directing correct forebrain development. An upstream regulator for prechordal plate development during mid-gastrulation and thus indirectly also of Shh secretion from this domain was found with ZIC2, analyzed in *Zic2* mutants in mice [21]. We found that *zic2a* expression was ceased in the ANP domain after induction of *bmp4*. In our paradigm we induce *bmp4* at 8.5 hpf, a timepoint later than mid-gastrulation [31]. However, we also found a mild and variable decrease in expression of *shha/b* in the anterior region, the prechordal plate. We can neither be sure nor rule out the possibility that the loss of *zic2a* is causing the reduced levels of *shha/b* in the prechordal plate, even if the onset of *bmp4* induction is later than mid-gastrulation. Alternatively, the induction of *bmp4* could have affected the expression of *zic2a* and *shha/b* independently. Our findings indicate, however, that BMP antagonists are important for *zic2a* and *shha*/b induction or maintenance in the ANP region. It was shown before that the expression of *six3b* was depending on Hh signaling [22]. In our analysis, the expression of *six3b* was not lost after *bmp4* induction, but was condensed at the midline. This suggests that the level of *shha/b* was sufficiently high for *six3b* induction. Zic2 was also found to act downstream of Shh to affect *six3* expression in the forebrain in zebrafish [22]. This effect was, however, considerably later, at mid-somitogenesis, and influenced the development of the prethalamus [22].

We further found that *rx3* and *foxg1a* expression were lost from the eye field and the future telencephalon, respectively. The loss of *foxg1a* well explains the growth defects of the telencephalon afterwards, which we noticed at 24 hpf. Astonishing, however, was the finding of a ceased *rx3* expression, because at 24 hpf we found a condensed domain of *rx2* and *pax6a*-positive tissue in the dysmorphic forebrain, the crypt-oculoid. In anophthalmic zebrafish *rx3* mutants (*chokh*), *rx2* expression was shown to be depending on *rx3* [18]. Our own analysis of *rx3* Crispants from this study is well in line with this. In the anophthalmic medaka *rx3* mutants (*eyeless, el*), however, *rx2* expression could be detected [19]. This difference may be due to the difference in species, but also the nature of the mutation could play an important role. In the zebrafish *chokh* mutant a point mutation (s399) results in a premature stop and thus in a truncated *rx3*. In medaka *eyeless*, it is an intronic insertion that results in a transcriptional repression, which is also temperature-sensitive. In our *bmp4* induction paradigm we do not mutate the *rx3* locus, but rather negatively regulate the expression of *rx3*. It is conceivable that *rx3* is expressed at a level which is not detectable with WMISH but sufficient to induce *rx2* expression in our paradigm and maybe also in the medaka eyeless embryos. In both, the level of *rx3* was, nevertheless, not sufficient to facilitate eye field splitting and optic vesicle out-pocketing (Rembold et al., 2006, this study). Irrespectively of the exact level of *rx3* expression/repression, we detected a dramatic reduction in *cxcr4a* expression within the eye field after *bmp4* induction. *Cxcr4* was show to act downstream of *rx3* and together with *sdf1b* from the surface mesoderm to facilitate the evagination of the eye field [3]. The “loss” of *rx3* and subsequently of *cxcr4a* in our *bmp4* induction paradigm, thus, very likely explains the lack of optic vesicle out-pocketing. Bearing in mind also our previous findings [13,15], our data indicate that BMP antagonism is continuously important for morphogenetic movements during optic cup development starting with optic vesicle evagination. This also demonstrates nicely the link between “morphogenetic coloboma” that we observed previously and the HPE phenotype from this analysis.

Moreover, investigating the arrangement of the forebrain at 24 hpf in succession of an induction of *bmp4*, we found that the dysmorphic forebrain is flatter than in controls and the domains were arranged in a more linear way. This could be explained in part by the failure in optic vesicle out-pocketing. However, we also found that the hypothalamic subduction, normally occurring during neurulation, must have been hampered. The anterior ventral hypothalamic domain did not reach its position, but rather remained posterior to the crypt-oculoid. The subduction movement was shown to be important for eye field separation [4] and our data suggest that BMP antagonism is also important to facilitate this process.

## 4. Materials and Methods

### 4.1. Zebrafish Care

Zebrafish were kept in accordance with local animal welfare law and with the permit 35-9185.64/1.1 from Regierungspräsidium Freiburg. Fish were maintained in a constant recirculating system at 28 °C on a 12 h light: 12 h dark cycle. The following transgenic lines were used: *tg(hsp70l:bmp4, myl7:eGFP)* [14] *tg(Ola.rx2:bmp4, myl7:eGFP)* [13] *tg(BRE-AAVmlp:eGFP)* [15,32]. Zebrafish embryos were grown at 28 °C in petri dishes in zebrafish medium, consisting of 0.3 g/L sea salt in deionized water. If melanin-based pigmentation needed to be inhibited for downstream applications, embryos were grown in 0.2 mM phenylthiourea.

#### 4.1.1. Heat Shock Procedures

For induction of heat-shock inducible transgenes, embryos were transferred to 1.5 mL reaction tubes and incubated at 37 °C in a heating block (Eppendorf Thermomixer). The onset and the duration of the respective heat shocks varied details are given in the results.

#### 4.1.2. Laser Scanning Confocal Microscopy

Confocal images were recorded with an inverted TCS SP8 microscope (Leica). Embryos were embedded in 1% low-melting agarose (Roth) in glass-bottom dishes (MatTek). Live embryos were anaesthetized with MS-222 (Tricaine, Sigma-Aldrich) for imaging. Image stacks were recorded with a z-spacing of 3 µm, unless specified otherwise.

#### 4.1.3. Image Processing

Images from microscopy were edited for presentation using ImageJ (Fiji) software [33].

#### 4.1.4. In Situ hybridization

Whole-mount ISH was performed according to an established protocol [34].

The following probes were created cloning free, according to [35]: Chrd

The following primers/probes were used:Emx3 F5′-GAAGTGCTTCACGATTGAATC-3′; R 5′-TGAAATGACGTCAATGTCCTC-3′Fgf8a F:5′-GACTCATACCTTCACGGTTGAG-3′; R: 5′-TGCGTTTAGTCCGTCTGTTG-3′Foxg1a F:5′-ATGTTGGATATGGGAGAAAG-3′; R: 5′-AAGAAATAACTGGTCTGACC-3′

Fsta see Knickmeyer et al. 2018

Grem2b see Knickmeyer et al. 2018

Nog2: For:5′-ATGGGCAGCATCACCCG-3′ Rev: 5′- TCAGCACGAGCACTTGCA-3′Chrd: For:5′-TTGTATGGCAGCAGGCGTAT-3′ Rev: 5′-TTGTATGGCAGCAGGCGTAT-3′Her13 F:5′-CCACGCTGCTGAACTTAGAAA-3′; R: 5′-TCATCCAGGTCAGAGCAGAGA-3′

Pax2a see Eckert et al. 2019

Pax6a F:5′-AGATGGTTGCCAACAGTCAG-3′; R: 5′-GGGACATGTCTGGTTCACTG-3′Rx2 F:5′-GCCTCTCCACAGAAAGCTAC-3′; R: 5′-CGATACTAGAACTGCGGTCG-3′Rx3 F:5′-ATGAGGCTTGTTGGATCTCAG-3′; R: 5′-ATGAGGCTTGTTGGATCTCAG-3′

Shha see Knickmeyer et al. 2021

Shhb see Knickmeyer et al. 2021

Six3b F:5′-TTTGGTCGTTGCCCGTAGCACC-3′; R: 5′-CATCGAAATCAGAGTCACTGTC-3′Zic2a F:5′-ATTAAGCAAGAGCTCATCTG-3′; R: 5′-AACTGTGGACCGCTGAGGAAG-3′

#### 4.1.5. CRISPR/Cas9 F0 Analysis (Crispants)

Embryos in 1-cell stage were microinjected with 1 µM Cas9 protein (Alt-R S.p. Cas9 Nuclease V3, 1081059, Integrated DNA Technologies) and 1µg/µL sgRNA mix as described [20]. sgRNAs were designed using CCTop (http://crispr.cos.uni-heidelberg.de (accessed on 18 April 2023)) [36]. Sequences for sgRNAs:*rx3* T1:CCCGGCGTTTCCATATGGAT*rx3* T2:TGAACGTGGTTCGGTTCCGC*rx3* T3:CTTCGAGAAGTCGCACTATC*rx3* T4:GAGATGGGGCCGGTCAACCA

## 5. Conclusions

Here we show that BMP antagonism is important for different aspects of forebrain and early eye development. We found at least four BMP antagonists expressed in the ANP domain during ANP cleavage. Induced *bmp4* expression affects various aspects of ANP development, e.g., expression of *shha/b* in the prechordal plate, expression of *zic2a* in the ANP, expression of *foxg1a*, *rx3* and *cxcr4a* in the ANP, respectively. Moreover, the hypothalamic subduction and neural keel formation was hampered. Together these changes result in anophthalmia with a crypt-oculoid in a dysmorphic forebrain. It will be interesting in future analyses to address further the link between the *bmp4* induction, the regulation of *shha/b* and *zic2a* and the subduction movement of the hypothalamic domain. Further, it will be interesting to address why the splitting defect of the ANP resulted in anophthalmia with a crypt-oculoid and not in cyclopia.

## Figures and Tables

**Figure 1 ijms-24-08052-f001:**
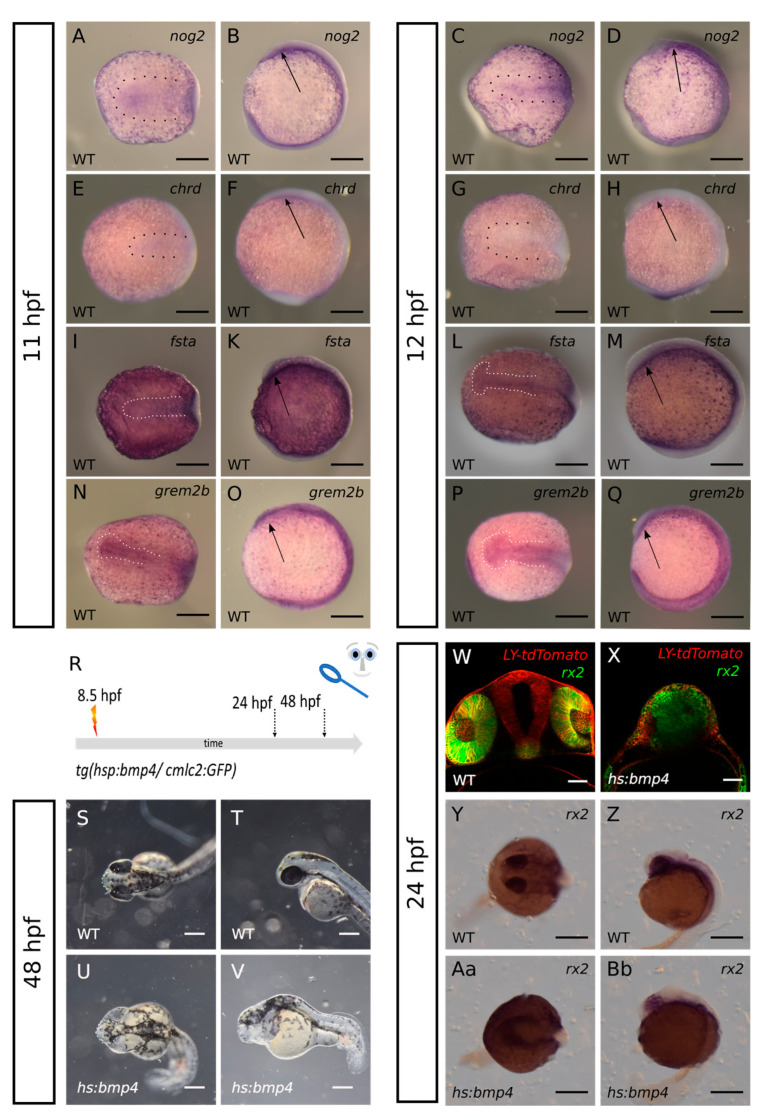
(**A**–**Q**): expression of bmp-antagonists *nog2*, *chrd*, *fsta* and *grem2b* in the ANP at 11 and 12 hpf. (**A**–**D**): ISH for *nog2*. (**A**,**B**): 11 hpf; (**C**,**D**): 12 hpf. (**E**–**H**): ISH for *chrd*. (**E**,**F**): 11 hpf; (**G**,**H**): 12 hpf. (**I**–**M**): ISH for *fsta*. (**I**,**K**): 11 hpf; (**L**,**M**): 12 hpf. (**N**–**Q**): ISH for *grem2b*. (**N**,**O**): 11 hpf; (**P**,**Q**): 12 hpf. (**A**,**C**,**E**,**G**,**I**,**L**,**N**,**P**): dorsal view. (**B**,**D**,**F**,**H**,**K**,**M**,**O**,**Q**): lateral view. Scalebars indicate 200 µm. (**R**): summary of experimental procedure. (**S**–**Bb**): Embryos were heat shocked at 8.5 hpf and observed at 24 hpf or 48 hpf, resp. (**S**,**T**): bright-field image of wildtype at 48 hpf. (**U**,**V**): bmp4-induced larva at 48 hpf. (**S**,**U**): dorsal view. (**T**,**V**): lateral view. Scalebars indicate 250 µm. (**W**,**X**): transversal confocal sections of larvae with rx2:GFP at 24 hpf. injection of *LY-tdTomato* mRNA in zygote. (**W**): wildtype. (**X**): bmp4-induced, scalebars indicate 50 µm. (**Y**–**Bb**): ISH for *rx2* at 24 hpf. (**Y**,**Z**): wildtype. (**Aa**,**Bb**): bmp4-induced. (**Y**,**Aa**): dorsal view; (**Z**,**Bb**): lateral view. Scalebars indicate 250 µm.

**Figure 2 ijms-24-08052-f002:**
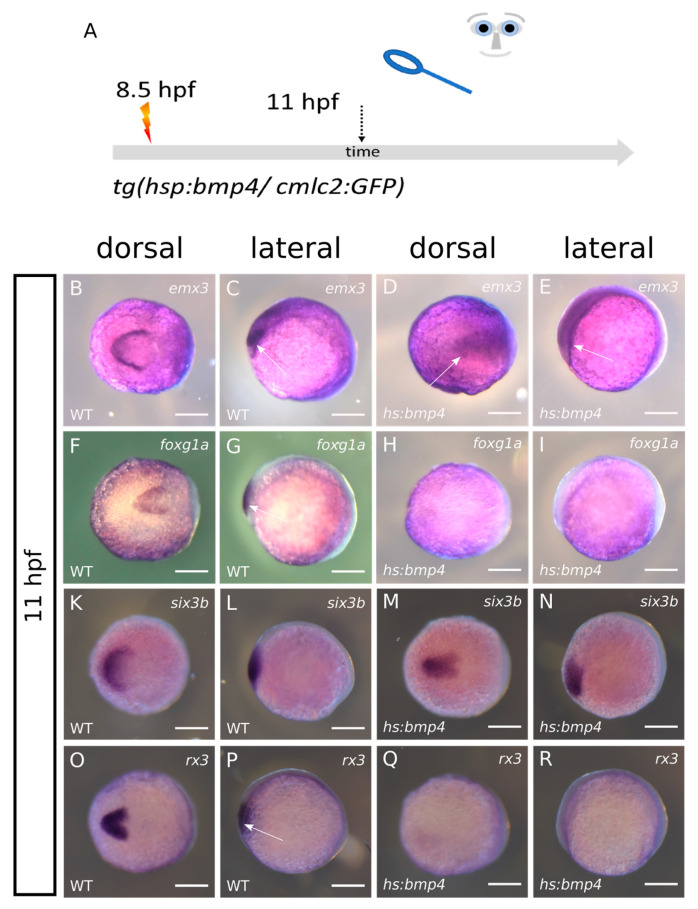
ISH at 11 hpf after bmp4-induction at 8.5 hpf. left columns wildtypes, right columns *bmp4*-induced. first and third column dorsal view, second and fourth column lateral view. (**A**): summary of experimental procedure. Embryos were heat-shocked at 8.5 hpf and analyzed at 11 hpf. (**B**–**E**): Expression of *emx3* is condensed at the midline after *bmp4*-induction (arrows). (**F**–**I**): the *foxg1a* domain in the forebrain (arrow) is lost after *bmp4*-induction. (**K**–**N**): expression of eye-field marker *six3b* is also condensed at the midline after *bmp4*-induction. (**O**–**R**): expression of eye-field marker *rx3* (arrow) is lost after *bmp4*-induction. Scalebars indicate 200 µm.

**Figure 3 ijms-24-08052-f003:**
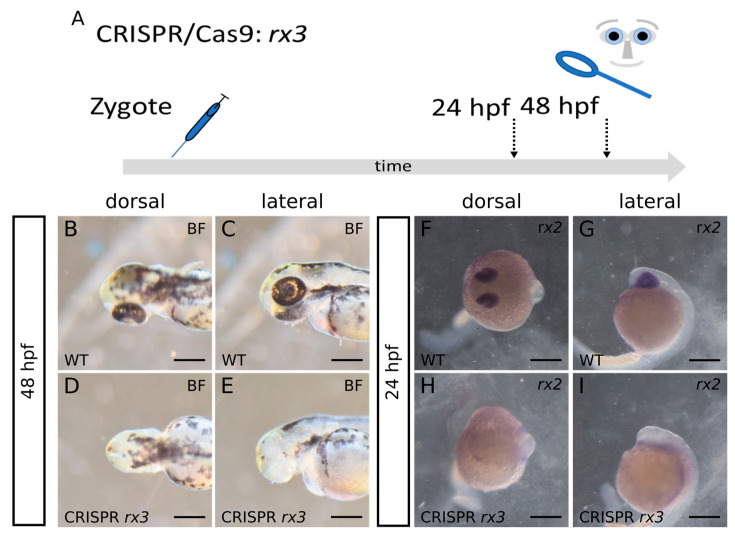
(**A**): summary of experimental procedure: embryos were injected at 1-cell-stage and analyzed at 24 hpf or 48 hpf. (**B**–**E**): Brightfield-images at 48 hpf. We observed anophthalmia in F0 *rx3* Crispants. (**F**–**I**): ISH for *rx2* at 24 hpf. The expression of *rx2* is absent in F0 *rx3* Crispants at 24 hpf, correlating with the severity of the phenotype. (**B**,**C**,**F**,**G**): wildtype. (**D**,**E**,**H**,**I**): *rx3*-crispant. (**B**,**D**,**F**,**H**) dorsal view. (**C**,**E**,**G**,**I**) lateral view. scalebars indicate 250 µm.

**Figure 4 ijms-24-08052-f004:**
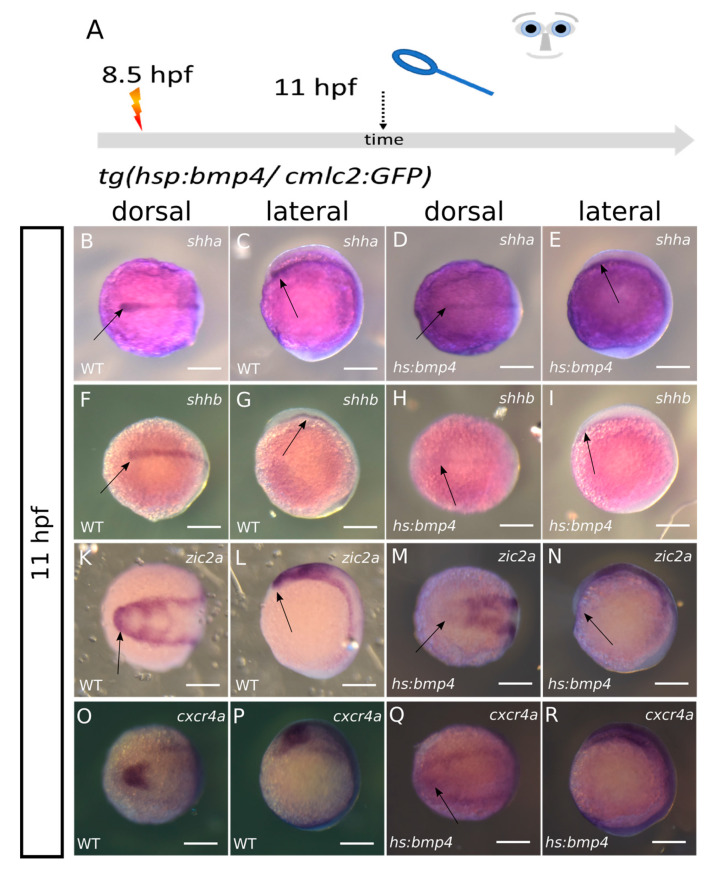
ISH at 11 hpf after bmp4-induction at 8.5 hpf. (**A**): summary of experimental procedure: embryos were heat-shocked at 8.5 hpf and analyzed at 11 hpf; left columns wildtypes, right columns *bmp4*-induced;first and third column dorsal view, second and fourth column lateral view. (**B**–**E**): Expression of *shha* is reduced in the area of the prechordal plate (arrows) after *bmp4*-induction yet still present. (**F**–**I**): *shhb* expression is also still present after *bmp4*-induction but reduced in intensity (arrows). (**K**–**N**): Expression of *zic2a* is absent in the ANP domain (arrows) after *bmp4*-induction. (**O**–**R**): expression of *cxcr4a* in the eye field is strongly reduced after *bmp4*-induction. scalebars indicate 200 µm.

**Figure 5 ijms-24-08052-f005:**
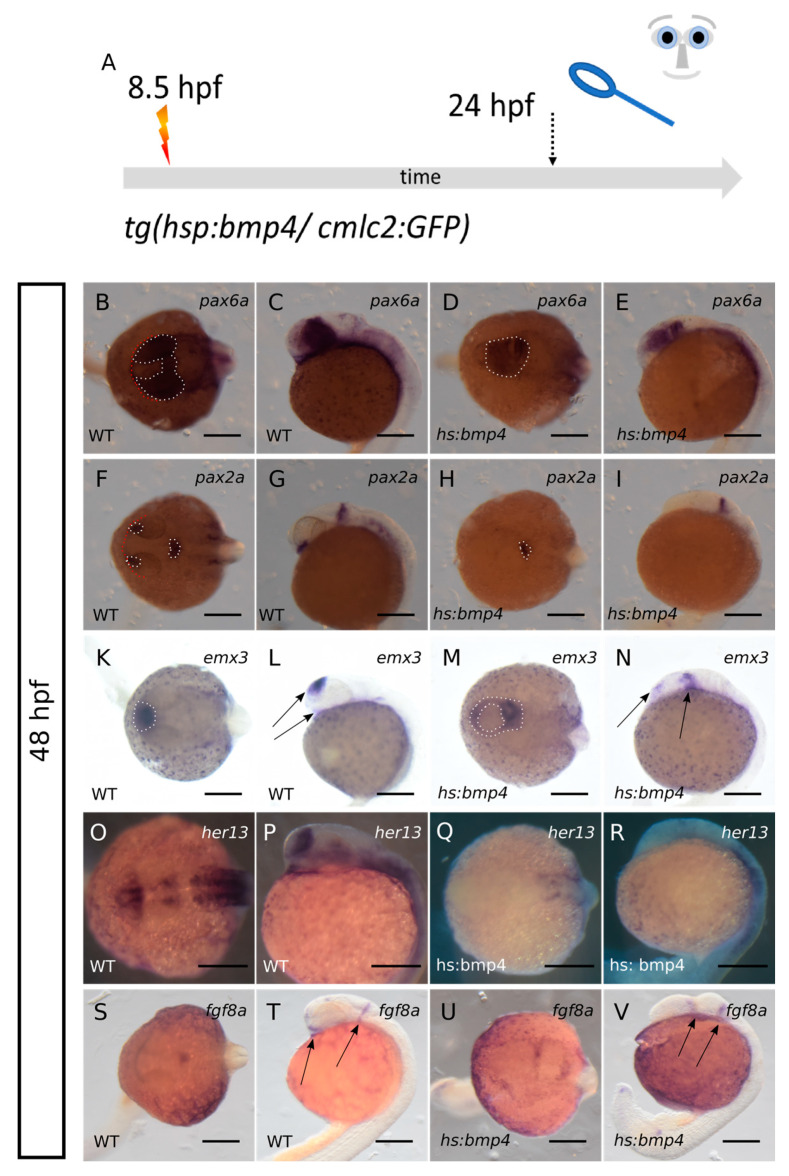
Analysis of marker genes of telencephalic, retinal and hypothalamic precursors after *bmp4*-induction. (**A**): summary of experimental procedure: embryos were heat-shocked at 8.5 hpf and analyzed at 24 hpf. Left columns wildtypes, right columns *bmp4*-induced embryos. First and third column dorsal view, second and fourth column lateral view. (**B**–**E**): ISH for *pax6a* at 24 hpf. *Pax6a* (dotted lines) is expressed in the crypt-oculoid and in the diencephalic domain after *bmp4*-induction. (**F**–**I**): ISH for *pax2a* (dotted lines) at 24 hpf. The expression of *pax2a* in the MHB is conserved after *bmp4*-induction but the optic stalk domain is lost. (**K**–**N**): ISH for *emx3* at 24 hpf. *Emx3* is present after *bmp4*-induction but the expression pattern is changed (arrows). (**O**–**R**): ISH for *her13* at 24 hpf. The expression of *her13* is lost in anterior regions after *bmp4*-induction. (**S**–**V**): ISH for *fgf8a* at 24 hpf. Two expression domains posterior to the crypt-oculoid are visible after *bmp4*-induction (arrows). The ventral expression domain (left arrow in **T**) is ceased after *bmp4*-induction. Scalebars indicate 200 µm.

**Figure 6 ijms-24-08052-f006:**
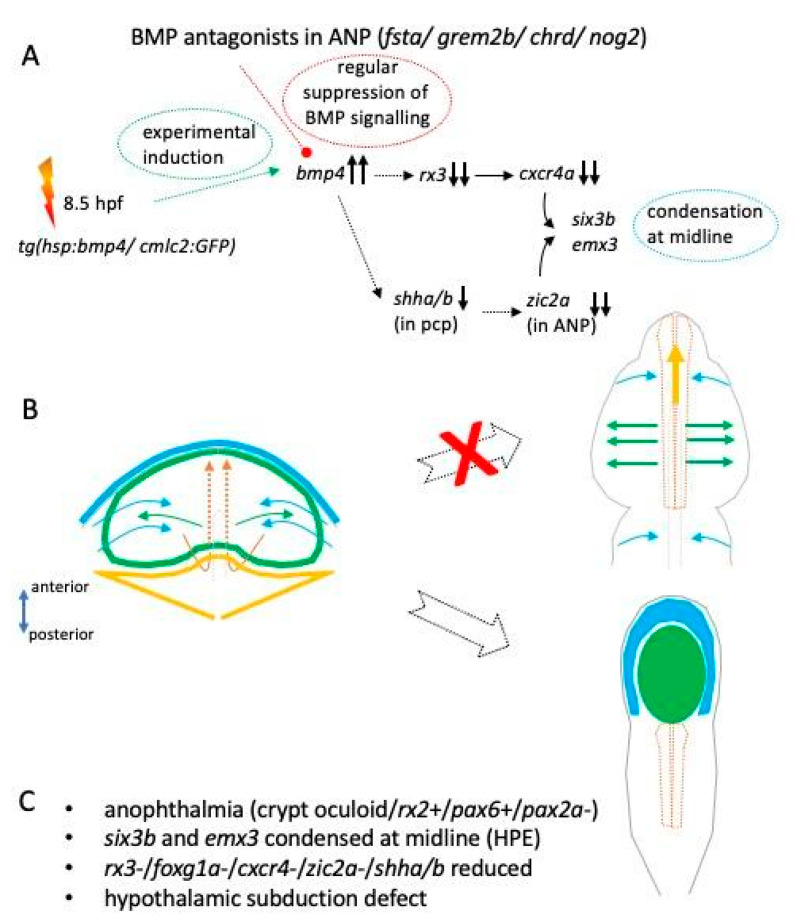
Scheme, summary of major findings. (**A**): Experimental induction of *bmp4* results in a suppression of *rx3* and *shha/b* with a consecutive suppression of *cxcr4a* and *zic2a*. The ANP is not splitting in a proper manner, resulting in a crypt-oculoid. *Six3b* and *emx3* are found condensed at the midline. (**B**): scheme showing the pathological development of the ANP resulting from *bmp4* induction, including a defect in the hypothalamic subduction. (**C**): brief summary of findings.

## Data Availability

No additional data is published with this article.

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
