# Peer review of "Holoprosencephaly with a Special Form of Anophthalmia Result from Experimental Induction of bmp4, Oversaturating BMP Antagonists in Zebrafish"

_ijms, 2023, doi:10.3390/ijms24098052_

Round 1
Reviewer 1 Report
This manuscript reports the role of bone morphogenetic proteins (BMPs) antagonism during anterior neural plate (ANP) and eye field development, aiming at the understanding on holoprosencephaly (HPE). Authors experimentally induced bmp4 expression at 8.5 hpf, resulting in anophthalmia. It was found that rx2 and pax6 positive retinal progenitor cells being stuck in the forebrain. Division of the telencephalic field and the division of the eye field were hampered by the expression of markers within the ANP at 11 hpf. A loss of rx3 and cxcr4 expression, a reduced expression of shha/b in the prechordal plate and a ceased zic2 expression in the ANP was found. This work identified a dysmorphic forebrain containing retinal progenitors, and the term crypt-oculoid was coined for this effect. Experiments were well carried out, step by step. In general, this manuscript is well written.
I do not have comments, except the new technical term. Since the phenotype “crypt-oculoid” is new and termed in this work, so it is not widely used. Therefore, I recommend not to put the word “crypt-oculoid” in the title.
Moreover, future work or prospective should be provided at the end of the manuscript.
English seems to be suitable.
Author Response
Ad Referee 1:
We thank the referee for the work and for the supporting words.
Please find enclosed a “point by point” rebuttal.
I do not have comments, except the new technical term. Since the phenotype “crypt-oculoid” is new and termed in this work, so it is not widely used. Therefore, I recommend not to put the word “crypt-oculoid” in the title.
We thank the referee for the recommendation. We changed the title accordingly and put the new term to the key words. Please see the revised version of our manuscript.
Moreover, future work or prospective should be provided at the end of the manuscript.
In the IJMS template form there is a section for the “conclusion” (5.). There, we placed a brief perspective. We extended this outlook. Please see the revised version of our manuscript.
Reviewer 2 Report
Dear authors,
I will recommend your manuscript for publication, but I has some questions for edition.
L12 – in abstract too much information not connected to results of research.
L26 – “BMP, BMP antagonists”- not correct keywords
L47 – add “Sonic hedgehog proteins”
L73 – add in paragraph info about object of study – zebrafish.
L75 – Not found what is “hpf”. Hours post fertilization?
L86 – Time line in picture wrong. 8.5, 24 and 48. But in photos is 11, 12, 24 and 48. Remove it or edit.
L119 – Not understanding, why only one figure moved to supplementary?
L291 – only here showing number of investigated embryos. It is necessary add in all parts of experiment.
L331 – Wrong footnote. Describe figure.
L336 – add reference for last sentence.
L417, 422 – Is it you’re methods? Else add reference.
L465 – Did you’re mean sgRNA sequences?
Regards,
Author Response
Ad Referee 2:
We thank the referee for the work and for the constructive comments on our work.
Please find enclosed a “point by point” rebuttal.
Dear authors,
I will recommend your manuscript for publication, but I has some questions for edition.
L12 – in abstract too much information not connected to results of research.
We thank the referee for this feedback. We intended to give a broader audience the possibility to connect to our work. Therefore, we started with general aspects and did not dive into the data at the beginning of the abstract. We would like to keep this version of the abstract.
L26 – “BMP, BMP antagonists”- not correct keywords
We changed this to “Bone morphogenetic protein (BMP)” and “antagonists for Bone morphogenetic protein (BMP antagonists)”
L47 – add “Sonic hedgehog proteins”
We changed it to “…is the Sonic hedgehog protein (Shh),...“
L73 – add in paragraph info about object of study – zebrafish.
We added the object of study in this paragraph. Please see the revised version of our manuscript.
L75 – Not found what is “hpf”. Hours post fertilization?
We thank the referee for the feedback. Yes, indeed this is the abbreviation for hours post fertilization. We now introduced the term in the text.
L86 – Time line in picture wrong. 8.5, 24 and 48. But in photos is 11, 12, 24 and 48. Remove it or edit.
We are sorry for the misunderstanding. The timeline is meant for the experimental setup from “S” to “Bb” and is thus correct. The ISH presented in in “A-Q” were performed on wildtype embryos. We added information to the figure legend to explain it better.
L119 – Not understanding, why only one figure moved to supplementary?
The data presented here is providing technical insights which is not adding information to the data of the figure. In our opinion the other figures provide data which is important for the analysis. Thus, the other data is presented in main figures.
L291 – only here showing number of investigated embryos. It is necessary add in all parts of experiment.
We provided this information in other aspects of this as well as for other experiments, e.g. line 124, 126, 137…
L331 – Wrong footnote. Describe figure.
The respective figure is the summary of major findings and is already providing the description. A long description in the legend would be highly redundant to the content of the figure as well as to the summary section. We are nevertheless grateful for the comment and added a brief legend to the figure.
L336 – add reference for last sentence.
We included references. Please see the revised version of our manuscript.
L417, 422 – Is it you’re methods? Else add reference.
Indeed, this is our methods, however, these are more or less standard procedures.
L465 – Did you’re mean sgRNA sequences?
We are sorry for not stating this aspect correctly. Sure, we were referring to the sgRNAs. We changed this aspect. Please see the revised version of our manuscript.